# A Secreted Lignin Peroxidase Required for Fungal Growth and Virulence and Related to Plant Immune Response

**DOI:** 10.3390/ijms23116066

**Published:** 2022-05-28

**Authors:** Feng Xiao, Wenxing Xu, Ni Hong, Liping Wang, Yongle Zhang, Guoping Wang

**Affiliations:** 1College of Plant Science and Technology, Huazhong Agricultural University, Wuhan 430070, China; fengxiao@webmail.hzau.edu.cn (F.X.); xuwenxing@mail.hzau.edu.cn (W.X.); whni@mail.hzau.edu.cn (N.H.); wlp09@mail.hzau.edu.cn (L.W.); zhangyongle@webmail.hzau.edu.cn (Y.Z.); 2Key Laboratory of Plant Pathology of Hubei Province, Wuhan 430070, China; 3Key Laboratory of Horticultural Crop (Fruit Trees) Biology, and Germplasm Creation of the Ministry of Agriculture, Huazhong Agricultural University, Wuhan 430070, China; 4Hubei Hongshan Laboratory, Wuhan 430070, China

**Keywords:** *Botryosphaeria*, lignin peroxidase, secreted protein, virulence, plant immunity, MAMP

## Abstract

*Botryosphaeria* spp. are important phytopathogenic fungi that infect a wide range of woody plants, resulting in big losses worldwide each year. However, their pathogenetic mechanisms and the related virulence factors are rarely addressed. In this study, seven lignin peroxidase (LiP) paralogs were detected in *Botryosphaeria kuwatsukai*, named BkLiP1 to BkLiP7, respectively, while only BkLiP1 was identified as responsible for the vegetative growth and virulence of *B. kuwatsukai* as assessed in combination with knock-out, complementation, and overexpression approaches. Moreover, BkLiP1, with the aid of a signal peptide (SP), is translocated onto the cell wall of *B. kuwatsukai* and secreted into the apoplast space of plant cells as expressed in the leaves of *Nicotiana benthamiana*, which can behave as a microbe-associated molecular pattern (MAMP) to trigger the defense response of plants, including cell death, reactive oxygen species (ROS) burst, callose deposition, and immunity-related genes up-regulated. It supports the conclusion that BkLiP1 plays an important role in the virulence and vegetative growth of *B. kuwatsukai* and alternatively behaves as an MAMP to induce plant cell death used for the fungal version, which contributes to a better understanding of the pathogenetic mechanism of *Botryosphaeria* fungi.

## 1. Introduction

The genus *Botryosphaeria* contains several pathogenetic species that cause severe diseases in many important woody plants worldwide [1,2]. Among *Botryosphaeria* species, *Botryosphaeria kuwatsukai* and the related species always infect pear, apple, and other pome fruit trees, cause ring rot disease [3,4], leading to symptoms such as stem wart, stem canker, branch dieback, and fruit rot, and result in significant yield losses and economic damage to these fruit trees [1,5,6]. E.g., *B. obtusa* resulted in 25–50% losses of apple fruits in southeastern United States in 1924, and *B. dothidea* caused losses of as high as 100% in the same region in 1954. However, the pathogenetic mechanisms of *Botryosphaeria* fungi are rarely addressed, and in particular, the virulence factors remain less determined, partly due to their multi-nuclei nature leading to a strong difficulty to establish a system for complete gene knock-out [7].

Genomics and transcriptomics analyses indicate that *B. kuwatsukai* contains many genes encoding carbohydrate-active enzymes (CAZYs) [8,9]. Currently, there are five main categories of CAZYs, including glycosyltransferases (GTs), glycoside hydrolases (GHs), polysaccharide lyases (PLs), carbohydrate esterases (CEs), and auxiliary activities (AAs) [10]. Most CAZYs are known as cell-wall-degrading enzymes (CWDEs), which play central roles in plant cell wall decomposition by fungi and bacteria [11]. As the penetration of the plant cell wall is an essential step to facilitate colonization in host tissue for all phytopathogens, CWDEs have been identified as important virulence factors in some fungi [12], e.g., *Botrytis cinerea* [13,14], *Fusarium oxysporum* [15], *Lasiodiplodia theobromae* [4,16], *Magnaporthe oryzae* [17], *Phytophthora parasitica* [18], *Sclerotinia sclerotiorum* [19], *Valsa mali* [20], and *Verticillium dahliae* [21]. However, the CAZYs in *Botryosphaeria* remain undetermined, and their functions are still unclear, except that the cutinase Bdo_10846 belonging to CWDEs was characterized as a virulence factor in *B. dothidea* [22].

To combat fungal infection or other kinds of pathogens, plants have developed two sophisticated defense lines: microbe-associated molecular pattern (MAMP)-triggered immunity (PTI) and effector-triggered immunity (ETI) [23,24]. MAMPs are conserved molecules present in microbes but absent in host plants, e.g., flagellin flg22 in *Pseudomonas syringae* [25,26] and EF-Tu in *Escherichia coli* [27,28]. In addition, some CWDEs secreted by phytopathogens can be recognized by plants, which also acted as MAMPs [29], e.g., ethylene-inducing xylanase (Eix) in *Trichoderma viride* [30], glycoside hydrolase family 12 (GH12) protein (XEG1) in *Phytophthora sojae* [31], pectate lyase VdPEL1, cutinase VdCUT11 in *V. dahliae* [21,32], and xylanase BcXyl1 in *B. cinerea* [13]. Correspondingly, plants activate a series of immune responses, including programmed cell death (PCD), ROS burst, callose deposition, and expression of immunity-related genes [33,34], to fend off the invasion of phytopathogens. Hypersensitive response (HR), a type of PCD, is a defense mechanism against biotrophic pathogens, which can, however, help the invasion in case of infection by necrotrophic pathogens [35,36]. Despite the economic significance of *Botryosphaeria* fungi, the majority of genes from these fungi have not been functionally characterized, particularly their roles involved in plant interaction, except for a limited function identification for cutinase Bdo_10846, two candidate effectors, and autophagy-related gene (BdATG8) in *B. dothidea* [22,37,38].

Lignin peroxidases (LiPs, EC 1.11.1.14) are the main lignin-degrading enzyme in the process of lignin biodegradation, which can oxidize phenolic and nonphenolic structures of lignin directly [39,40]. They belong to the auxiliary activities family 2 (AA2), which were first described in *Phanerochaete chrysosporium* [41]. However, most of them are focused on the biochemical function of enzyme activity, e.g., *P. chrysosporium* [42,43,44], *Pseudomonas aeruginosa*, and *Serratia marcescens* [45]. Among ligninolytic enzymes, only a few characterized as virulence factors, including laccase (Lac) and manganese peroxidase (MnP), e.g., *Lac2*, a laccase gene that is involved in the appressorial melanization of *Colletotrichum orbiculare* [46], and MnP, have a function in fungal growth and development and stress response in *Trametes trogii* [47]. Meanwhile, LiP has been rarely studied [48]. Interestingly, the production of stem wart and stem canker symptoms may be related to the accumulation of lignin in pear trees as infected by *Botryosphaeria* fungi. Therefore, it is important to characterize LiP in *B. kuwatsukai* considering that it may be closely related to the pathogenetic mechanism of these.

In this study, seven fungal LiP paralogs belonging to AA2 were detected in *B. kuwatsukai*, whereas only BkLiP1 was identified as responsible for the fungal vegetative growth and virulence, as well as an MAMP that can trigger plant immune responses, which contributes to a better understanding of the pathogenetic mechanism of *Botryosphaeria* fungi.

## 2. Results

### 2.1. Seven Lignin Peroxidase Paralogs Detected in B. kuwatsukai

A total of seven paralogs containing the conserved domain of lignin peroxidase (LiP) were detected in the genome (accession no. SACU00000000) of *B. kuwatsukai* as searched with the conserved domain approach and serially named *B. kuwatsukai* Lignin Peroxidase 1 (BkLiP1) to 7 based on their position in the genome. The open reading frame (ORF) of *BkLiP1* (GenBank: ON184087) is 1116 base pairs (bp) encoding a 371 amino acid (aa) protein with a predicted N-terminal signal peptide (SP, 1–21 aa), and the ORF of *BkLiP2* (GenBank: ON184088) is 1048 bp with two introns, encoding a 312 aa protein with a predicted SP (1–19 aa) at the N-terminus. No transmembrane helice motifs are observed within both proteins, suggesting that they may be secreted into the extracellular space. Phylogenetic analysis showed that orthologs of BkLiP1 to BkLiP4, with SPs, were separately clustered with their orthologs in other fungi, while BkLiP5 to BkLiP7, without SPs, were phylogenetically related to ascorbate peroxidase 1 of *Arabidopsis thaliana* (Figure 1A).

To check the expression level in pear plants, *BkLiPs* were subjected to an assessment of their expression levels after being inoculated on the leaves, fruits, and shoots of pear (*Pyrus pyrifolia* cv. Hohsui) with the mycelia of *B. kuwatsukai*. Quantification by real-time quantitative reverse-transcription PCR (RT-qPCR) showed that the expression levels of *BkLiP1* and *BkLiP2* were significantly up-regulated, ranging from 10 to 35 folds higher on pear plants than those on potato dextrose agar (PDA) at 3 days post-inoculation (dpi), whereas the remaining *BkLiPs* remained consistent or slightly increased (Figure 1B). Moreover, as the inoculation time was prolonged, the expression level of *BkLiP1* was straightly increased, i.e., from 2 folds at 12 h post-inoculation (hpi) up to 22 folds at 72 hpi (Figure 1C), while the expression level of *BkLiP2* fluctuated from 4 folds at 12 hpi, up to 21 folds at 48 hpi, and 3 folds at 72 hpi (Figure 1D). These results indicate that *BkLiP1* and *BkLiP2* may be involved in the infection process of *B. kuwatsukai* on pear.

### 2.2. BkLiP1 Contributes to the Vegetative Growth and Virulence in B. kuwatsukai

To determine the roles in *B. kuwatsukai*, *BkLiP1* was knocked out from a wild-type (WT) strain SZ2l62 of *B. kuwatsukai*, generating three *BkLiP1*-deletion mutants (∆*BkLiP1-*8, ∆*BkLiP1-*21, and ∆*BkLiP1-*26) by the split-marker approach (Appendix A). Compared with the WT strain, three ∆*BkLiP1* mutants grew more slowly on PDA, with the colony diameters of 59.1 mm and smaller than that (68.1 mm) of the WT strain at 3 dpi, and produced few aerial mycelia (Figure 2A,B). In addition, ∆*BkLiP1* mutants induced significantly shorter lesions than those induced by the WT strain on pear leaves (Figure 2C), as well as on the shoots (Figure 2F) and fruits (Appendix A), suggesting that the virulence of ∆*BkLiP1* mutants was attenuated (Figure 2D,G and Appendix A). When assessed by qPCR with genomic DNA isolated from infected pear leaves at 4 dpi, fungal biomass was significantly reduced in pear leaves inoculated with the ∆*BkLiP1* mutants, in comparison to those inoculated with the WT strain (Figure 2E). Meanwhile, as *BkLiP1* was recovered, generating a complemented transformant (termed cBkLiP1) from a deletion mutant (∆*BkLiP1*-8) by the split-marker approach (Appendix A), it restored the growth (Figure 2A,B) and virulence (Figure 2C,D,F,G) to the level of the WT strain, as shown by that the colony diameter of the cBkLiP1 transformant was 68.0 mm similar to that (68.1 mm) of the WT strain at 3 dpi (Figure 2B), and the lesion lengths induced cBkLiP1 transformant was 8.5 mm similar to that (9.2 mm) by the WT strain (Figure 2G). In addition, the conidiation, hyphal separation, and mycelium weight were also restored for the cBkLiP1 transformant as compared with the WT strain (Appendix A). Furthermore, *BkLiP1* overexpression strains (OEBkLiP1-1 and OEBkLiP1-8) were also generated (Appendix A), and they exhibited no obvious changes in the morphology, vegetable growth (Figure 2A,B), and virulence on pear leaves (Figure 2C,D) but had significantly higher virulence on pear shoots (Figure 2F,G) as compared with those of the WT strain or *BkLiP1* deletion mutants. These results suggest that BkLiP1 contributes to the vegetative growth and virulence of *B. kuwatsukai*.

### 2.3. BkLiP2 Has No Obvious Impact on the Virulence in B. kuwatsukai

To determine the roles in *B. kuwatsukai*, *BkLiP2* was knocked out from the WT strain, generating three *BkLiP2*-deletion mutants (∆*BkLiP2*-34, ∆*BkLiP2*-39, and ∆*BkLiP2*-54) with the split-marker approach (Appendix A). Unlike ∆*BkLiP1* mutants, the colony morphology and vegetative growth were different among the ∆*BkLiP2* mutants, though they had also decreased mycelia growth in comparison to the WT strain (Appendix A). Nevertheless, no significant difference was observed in the virulence between the ∆*BkLiP2* mutants and the WT strain, either in the lesion length on the pear fruit surface or the lesion depth inside the fruit (Appendix A). These results indicate that BkLiP2 has no obvious impact on the virulence of *B. kuwatsukai*.

### 2.4. The SP in BkLiP1 Required for Its Secretion and Function

The bioinformatic analysis predicts that BkLiP1 contains an SP at the N-terminus (1–21 aa) while no transmembrane helice motif, suggesting that it may be secreted into the extracellular space. To confirm the speculation, the SP-encoding region was constructed into the vector pSUC2 and transformed into the yeast YTK12 [49]. The resulting pSUC2-BkLiP1SP transformant was able to grow well on the YPRAA agar medium and had secreted invertase activity, similar to the positive control pSUC2-Avr1bSP (Figure 3), suggesting that the SP has a secretion function.

To more strictly assess the role of SP in the function of BkLiP1, a complemented transformant without SP (termed cBkLiP1^∆SP^) was generated from ∆*BkLiP1*-8 mutant (Appendix A) and subjected to biological assessment. The resulting cBkLiP1 transformant was normal in virulence; however, the cBkLiP1^∆SP^ transformant, similar to the ∆*BkLiP1*-8 mutant, significantly decreased its virulence on pear leaves (Figure 2C,D) and shoots (Figure 2F,G), indicating that the SP is essential for the complementation of ∆*BkLiP1* mutants. Furthermore, the hyphal separation, conidiation, and mycelia dry weight of cBkLiP1^∆SP^ transformant also decreased (Appendix A), though their morphology (Figure 2A) and conidial germination rates stayed similar to the WT strain (Appendix A). Therefore, SP plays an important role in the secretion of BkLiP1, which functions in the virulence and vegetative growth of *B. kuwatsukai*.

### 2.5. BkLiP1 Localizes on the Cell Wall of B. kuwatsukai

To observe the localization of BkLiP1 in *B. kuwatsukai*, *BkLiP1* was fused to the N-terminus of an enhanced green fluorescent protein (eGFP) gene within a pCETNS4-eGFP vector and transformed into the WT strain, generating a BkLiP1-eGFP transformant. As checked under fluorescence microscopy, it revealed that green fluorescence accumulated on the cell walls of hyphae and conidia in the BkLiP1-eGFP transformant, which was further confirmed by the green fluorescence absent in the protoplasts of BkLiP1-eGFP transformants as their cell walls were removed with the treatment of lytic enzyme (Figure 4). In contrast, green fluorescence was uniformly distributed in the hyphae, conidia, and protoplast in the control strains transformed with the empty vector (WT-eGFP) (Figure 4). These results suggest that BkLiP1 localizes on the cell wall of *B. kuwatsukai*.

### 2.6. BkLiP1 Can Be Delivered into Plant Extracellular Space with the Aid of SP

To check whether BkLiP1 could be delivered into plant cells independently, nuclear targeting assay was used to facilitate visualization of translocation of BkLiP1 according to previous research [50]. A small nuclear localization signal (NLS) from simian virus large T-antigen [51] was added at the C-terminus of *BkLiP1* in the presence and absence of the SP-encoding region after they were fused with an mCherry fluorescent protein-encoding gene, resulting in BkLiP1-mCherry-NLS and BkLiP1^∆SP^-mCherry-NLS vectors. The constructed vectors together with the control vector (mCherry-NLS) were separately transformed into the WT strain, and the resulting transformants were inoculated on onion bulbs by mycelial plugs. At 48 hpi, the lower epidermal layers of onion bulbs infected with the transformants were checked under the confocal microscope. The confocal images showed that the mCherry fluorescence was observed inside the intercellular space of epidermal cells that were infected by the mycelia transformed with BkLiP1-mCherry-NLS, as well as in the cells that were uninfected by the mycelia but close to the infected (Appendix A). However, the mCherry fluorescence was only observed inside of mycelia instead of the epidermal cells as inoculated with the transformants of BkLiP1^∆SP^-mCherry-NLS, as well as the WT strain transformed with the empty vector (WT-mCherry-NLS) (Appendix A). These results suggest that BkLiP1 is secreted into plant extracellular space with the aid of SP.

To further check the subcellular location of BkLiP1 in plant cells, *BkLiP1* was inserted into a plant-expression vector containing an enhanced yellow fluorescent protein (eYFP)-encoding gene, generating a pCNF3-BkLiP1-eYFP vector, transformed into *Agrobacterium tumefaciens* and infiltrated into the leaves of *N. benthamiana*. Observation under a confocal microscope at 2 dpi indicated that BkLiP1 was mainly distributed around the cell membrane (Figure 5A). Further research found that BkLiP1 was gathered in apoplast space as observed after cell plasmolysis (Figure 5B). Therefore, these results further confirm that BkLiP1 is a secreted protein that can be transmitted into the apoplast space of plant cells.

### 2.7. BkLiP1 Induces Plant Immunity Responses

To further explore whether BkLiP1 can induce PCD, *BkLiP1* was constructed into a potato virus X (PVX) vector and transformed into *A. tumefaciens*, which was infiltrated into *N. benthamiana* leaves. At 7 dpi, PCD was observed on the leaves infiltrated with BkLiP1 (Figure 6A); the symptom was similar to those inoculated with Bax, a proapoptotic protein derived from a mouse involved as a positive control [52]. In contrast, no PCD reaction was observed for the leaves infiltrated with *A. tumefaciens* containing BkLiP1^∆SP^ or eGFP (Figure 6A). Furthermore, we also tested whether cell death could be induced in *N. benthamiana* leaves by infiltrating with *Agrobacterium* to express the BkLiP1 and BkLiP1^ΔSP^ through generating a plant expression vector (pCNF3-eYFP as control). Cell death was particularly evident in *N. benthamiana* leaves infiltrated with BkLiP1 rather than by BkLiP1^∆SP^ at 13 dpi (Figure 6B). These results indicate that BkLiP1 is required to be secreted to the extracellular space to induce cell death in *N. benthamiana*.

To determine whether the PCD triggered by BkLiP1 is associated with plant defense response, ROS accumulation and callose deposition were tested in the infiltrated *N. benthamiana* leaves. A significant accumulation of ROS and callose deposition were observed in the leaves infiltrated by BkLiP1 at 48 hpi, while a slight or no accumulation in the ones by BkLiP1^∆SP^ (Figure 6C). Correspondingly, two HR marker genes *NbHIN1* and *NbHSR203J*, as well as genes related to the hormone signaling pathways, e.g., *NbPR1a*, *NbPR2*, *NbPR4*, *NbLOX*, and *NbERF1*, were significantly up-regulated after being infiltrated by BkLiP1 revealed by RT-qPCR (Figure 6D,E and Appendix A). However, these genes, except for *NbERF1*, were not up-regulated by infiltration of BkLiP1^∆SP^ (Figure 6D,E and Appendix A). To further check whether BkLiP1 can trigger PTI response in plants, some PTI marker genes in *N. benthamiana*, including *NbCYP71D20*, *NbPTI5*, *NbACRE31*, *NbWRKY7*, and *NbWRKY8*, were selected for qualification by RT-qPCR. The results show that these genes were all significantly up-regulated in *N. benthamiana* after infiltration by BkLiP1, while not by BkLiP1^∆SP^ (Figure 6F and Appendix A). These results indicate that BkLiP1 can stimulate the PTI reaction, and it is an MAMP.

## 3. Discussion

*B. kuwatsukai* is a necrotrophic pathogenic fungus that causes stem wart and stem canker symptoms in the pear and apple [1,3,6]. According to genomic annotation, more than 600 genes for encoding carbohydrate-active enzymes have been proposed in the genome of *B. kuwatsukai* [1,8], while most of them remain underdetermined for the biological functions in the interaction between *B. kuwatsukai* and host [22], especially those genes belonging to the AA family [53,54]. Here, seven fungal LiP paralogs belonging to AA2 were detected in the *B. kuwatsukai* genome. However, only BkLiP1 was confirmed to be involved in the fungal pathogenicity, as well as in the vegetative growth. Moreover, BkLiP1, secreted by *B. kuwatsukai*, could trigger cell death and plant immunity by targeting plant apoplast space. To the best of our knowledge, this is the first report of an AA family protein, BkLiP1, secreted by *B. kuwatsukai*, being required for fungal growth and virulence and triggering plant immune response.

Increasing evidence suggests that secreted proteins play an important role in the infection and colonization of necrotrophic pathogens [55,56,57], and the function of these proteins requires endocytosis by pathogens [58]; some of them are related to CWDEs [14,59], and others are important effectors [37,57,60]. In this study, seven genes encoding LiPs, belonging to AA2 related to CWDEs in *B. kuwatsukai*, were chosen for expression analysis. However, only two genes (*BkLiP1* and *BkLiP2*) were selected for further characterization since they were generally detectable in other filamentous fungi and significantly up-regulated during infection of *B. kuwatsukai*, suggesting that both genes may play general and important roles in the infection or pathogenicity in *Botryosphaeria* fungi. In addition, only *BkLiP1* instead of *BkLiP2* was closely related to the pathogenicity revealed by gene knock-out, complementation, and overexpression approaches. A similar phenomenon has ever been observed in other phytopathogenic fungi, e.g., in *B. cinerea*, only one subtilisin-like protease *Bcser2*, instead of *Bcser1*, showed crucial roles in the sclerotial formation, conidiation, and virulence [61], while in *F. oxysporum*, both *pg1* and *pgx6* genes encoding polygalacturonases were required for the virulence [62]. Moreover, BkLiP1, as a secreted protein, is necessary to be delivered into plant extracellular space to play its function because the SP of *BkLiP1* not only is essential for the complementation of the ∆*BkLiP1* mutant but also has an influence on the virulence, which is similar to Osp24 in *F. graminearum* [63] and FoEG1 in *F**. oxysporum* [15]. Thus, our results confirm that BkLiPs play diverse roles in the virulence and vegetative growth of *B. kuwatsukai*, while BkLiP1 appears to be critical in these processes [15,63]. Whether BkLiPs deserve other biological roles remains to be further explored.

BkLiP1 was translocated into the plant apoplast with the aid of SP and triggered plant defense response, including cell death, ROS burst, callose deposition, and high expression of a series of resistance genes involved in HR and hormone signaling pathways. The apoplastic space among plant cells is like a battlefield where most significant interactions between plants and pathogens happen [64,65]. Most secreted proteins have been reported to play a significant function in plant apoplastic space, such as BcXYG1 from *B. cinerea* [14], FoEG1 from *F. oxysporum* [15], and RcCDI1 from *Rhynchosporium commune* [66], which can induce cell death with their full length. Our findings imply that apoplastic space is very important for the function of BkLiP1 in *B. kuwatsukai*, suggesting that BkLiP1 is an apoplastic cell-death-inducing protein [67]. However, BkLiP1 without its SP (BkLiP1^∆SP^) could also trigger cell death, but the effect was significantly reduced. We speculate that the function of secreted proteins is enhanced by SP for its localization depending on the fungal species, since Ss-Caf1, a secreted protein from *S. sclerotiorum*, does induce severe necrosis without its signal peptide in plants [68], while SsCP1 [57] and SsSSVP1 [56] from *S. sclerotiorum* and VmE02 from *V. mali* [69] can also induce cell death with or without their SP.

Most related secreted proteins have been reported in some necrotrophic pathogens as MAMPs, e.g., endopolygalacturonases (PGs) in *B. cinerea* [70], glycoside hydrolase family 28 endopolygalacturonase (LtEPG1) in *L. theobromae* [16], and small cysteine-rich protein (VmE02) in *V. mali* [69]. Furthermore, MAMPs are often conserved molecules required for the development and virulence of microbes, whereas they can also be sensed by pattern recognition receptors (PRRs) of the plant cell membrane to activate plant immunity responses [71,72]. Here, besides as a virulence factor and involvement in the growth of *B. kuwatsukai*, BkLiP1 could also behave as an MAMP to induce cell death, which is a hypersensitive response (HR) resulted from a kind of immune response of plants against biotrophic pathogens. Conversely, the cell death reaction might be helpful for the invasion in case of infection by necrotrophic pathogens [35,36]. Therefore, we speculate that BkLiP1, as an MAMP, stimulates the plant immunity response to induce cell death of host plants that can contribute to further infection of *B. kuwatsukai*.

In conclusion, our results suggest that BkLiP1 plays an important role in the virulence and vegetative growth of *B. kuwatsukai* and alternatively behaves as an MAMP to induce plant cell death used for the fungal version. Hence, our findings offer an important clue to help decipher the pathogenetic mechanism in *B. kuwatsukai*.

## 4. Materials and Methods

### 4.1. Culture Conditions and Fungal Transformation

The wild-type *B. kuwatsukai* strain SZ2l62 was cultured on PDA plates at 28 °C. Gene disruption mutants and their corresponding complement transformants, *BkLiP1* overexpression strains, and fluorescent-labeled strains were cultured on PDA plates amended with hygromycin B at 50 μg/mL or G418 at 50 μg/mL (Sigma-Aldrich, Shanghai, China). To assay the expression pattern of *BkLiP1*, the WT strain was cultured on PDA for 3 days, and mycelia were collected, then transferred to new PDA plates or inoculated on leaves, fruits, and shoots of pear (*Pyrus pyrifolia* cv. Hohsui) at 25 °C. Mycelia were harvested at 0, 6, 12, 24, 48, and 72 h post-inoculation (hpi) for nucleic acid extraction. PDA cultures grown at 28 °C were used for measuring growth rate and observing colony morphology or hyphal tips. Then the colonies scratched off aerial mycelia incubated at 28 °C under black light were used to induce conidiation. Pycnidia per medium and conidia per pycnidia were counted under the microscope directly or with hemocytometer after being cultured on PDA at 28 °C for 7 days. Mycelia dry weight was assayed with the mycelia collected from liquid potato dextrose medium (50 mL) inoculated with four mycelia plugs (diameter = 5 mm) for 4 days and dried under 60 °C for 6 h.

Gene replacement constructs were generated with the split-marker approach [57,73] and transformed into protoplasts of wild-type strain SZ2l62 as previously described [74]. For each *LiP* gene, at least three independent gene replacement transformants were identified. For complementation assays, the entire *BkLiP1* gene and *BkLiP1* without SP gene combined with the aminoglycoside phosphotransferase (aph) at the N-terminus were generated through the split-marker approach and transformed into the protoplasts of *∆BkLiP1*-8 mutant as previously described [57]. The plasmid pCETNS4 was used to construct the *BkLiP1* overexpression vector. The CDs fragment of *BkLiP1* without stop codon was amplified from cDNA of *B. kuwatsukai* SZ2l62 and ligated between the *Spe* I and *Kpn* I site. The generation of transformants was performed as previously described [74]. The deletion mutants and complemented transformants were identified by PCR and RT-qPCR. All the primers used in this study are described in Appendix A.

### 4.2. Bioinformatics Analyses

The genome and predicted protein sequences of *B. kuwatsukai* were downloaded from a genome sequence file. To predict the *LiP* genes, the protein sequence of *B. kuwatsukai* was used as queries to search against the predicted proteomes of *B. dothidea* and *B. kuwatsukai* by BLASTp. Sequences of the *B. kuwatsukai* proteins also were used to search against the genome sequences of *B. dothidea* and *B. kuwatsukai* by tBLASTn for possible genes that might not be predicted by automated annotation. Reference sequences of LiPs were retrieved from the NCBI GenBank database. Structural domains of candidate LiPs were predicted using the NCBI Conserved Domain Search Tool (https://www.ncbi.nlm.nih.gov/Structure/cdd/wrpsb.cgi) (3 September 2019). The signal peptide was predicted using the SignalP-5.0 online tool (https://services.healthtech.dtu.dk/service.php?SignalP-5.0) (7 September 2021). TMHMM Server 2.0 (http://www.cbs.dtu.dk/services/TMHMM/) (5 October 2020) and SMART MODE (http://smart.emblheidelberg.de/smart/change_mode.pl) (2 December 2020) were used for the prediction of transmembrane helices. The multiple-sequence alignment of BkLiP1 and its orthologs in Botryosphaeriaceae or other pathogens was generated using the ClustalX v. 2.0 program (European Molecular Biology Laboratory, Hinxton, Cambridgeshire, UK) [75]. The phylogenetic tree was constructed with MEGA 7.0 (Sudhir Kumar, Arizona State University, Knicks, AZ, USA) [76] using the maximum-likelihood method.

### 4.3. RNA Extraction and RT-qPCR

Total RNA was extracted using TRIpure Reagent (Invitrogen, Carlsbad, CA, USA) according to the manufacturer’s instructions. First-strand cDNA was synthesized from 2 μg of total RNA using the TransScript 5X All-In-One RT MasterMix with AccuRT Genomic DNA Removal Kit (Abm, VAN, Canada), followed by qPCR using the iTaq^TM^ Universal SYBR^®^ Green Supermix (Bio-Rad, Hercules, CA, USA). The genes’ 18S ribosome RNA (18S rRNA) in *B. kuwatsukai* [77], *Actin* in *P. pyrifolia* [78], and *NbActin* in *N. benthamiana* [79] were used as internal controls. Relative expression levels were determined using the 2^−∆∆Ct^ method with three independent biological replicates. The 2^−∆∆Ct^ value was used to evaluate the fold change of gene expression. Each experiment had three replicates to give the main value, and the standard deviations were generated. To quantify fungal biomass in infected plant tissues, genomic DNA was extracted from infected pear leaves sampled at 4 dpi and used for qPCR assays with primers specific for the *P. pyrifolia Actin* and *B. kuwatsukai* 18S rRNA [63].

### 4.4. Plant Growth and Virulence Assays

*Nicotiana benthamiana* used in these experiments was grown at 25 °C in growth chambers under a long-light-period (16 h: 8 h, light: dark) condition. Virulence tests of *B. kuwatsukai* strains (wild-type and transformants) were conducted on detached healthy pear leaves (*P*. *pyrifolia* cv. Hohsui), pear shoots (*P. pyrifolia* cv. Hohsui), and pear fruits (*P. pyrifolia* cv. Hohsui and *P. bretschneideri* cv. Huangguan) for at least five repeats with mycelia plugs (diameter = 5.0 mm) excited from the margins of actively growing colonies on PDA. All the experiments were repeated three times.

### 4.5. Assays for the Function of the SP of BkLiP1

The gene encoding predicted SP (21 aa) of *BkLiP1* was cloned into the pSUC2 vector [80] that carries the yeast *SUC2* gene, and its SP sequence was deleted. The resulting pSUC2-BkLiP1SP construct was transformed into the yeast SUC2 mutant YTK12 [81] and assayed for growth on CMD-W (0.17% yeast nitrogen base without AAs, 0.074% tryptophan dropout supplement, 2% sucrose, 0.1% glucose, and 2% agar) and YPRAA medium plate (1% yeast extract, 2% peptone, 2% raffinose, 2% agar, and 0.2 μg/mL antimycin A) [49]. Transformants of YTK12 carrying the empty pSUC2 vector or pSUC2-Avr1bSP [82] were used as the negative and positive controls, respectively. The principle of color change was that the invertase enzymatic activity was detected by the reduction of 2,3,5-Triphenyltetrazolium Chloride (TTC) to insoluble red-colored 1,3,5-triphenylformazan (TPF) [49].

### 4.6. Assay for the Location of the BkLiP1 in B. kuwatsukai

To generate the BkLiP1-mCherry-NLS and BkLiP1^∆SP^-mCherry-NLS fusion constructs, the pCETNS4-mCherry-NLS fusion construct was generated at first; the sequence of mCherry-NLS was amplified from H_2_B-mCherry [83] vector using primers (Appendix A) with *Kpn* I and *Sma* I sites. Then the CDs fragment of *BkLiP1* and *BkLiP1*^∆SP^ without stop codon was amplified from cDNA of *B. kuwatsukai* SZ2l62 and ligated into the N-terminus of the mCherry at the *Spe* I and *Kpn* I site, respectively. The plasmid pCETNS4 was also used to construct the BkLiP1-eGFP vector. The eGFP fragment was amplified from PVX-eGFP with primers containing *Kpn* I and *Sma* I sites, then ligated into *Kpn* I and *Sma* I sites of pCETNS4 to generate pCETNS4-eGFP vector. The CDs fragment of *BkLiP1* without stop codon was ligated into the N-terminus of the eGFP at the *Spe* I and *Kpn* I site. The BkLiP1-mCherry-NLS, BkLiP1^∆SP^-mCherry-NLS, and BkLiP1-eGFP fusion constructs were transformed into the wild-type strain SZ2l62 using the PEG-mediated transformation method [74]. Tissues from onion bulb lower epidermal cells infected with the *B. kuwatsukai* engineered strains expressing BkLiP1-mCherry-NLS and BkLiP1^∆SP^-mCherry-NLS fusion proteins were examined under confocal laser scanning microscopy (CLSM; Leica Microsystems, TCS-SP8, Germany) at 48 h post infiltration, respectively. The mycelia, conidia, and protoplasts of the *B. kuwatsukai* engineered strains were viewed under a fluorescence microscope (CLSM; Leica Microsystems, DM2500, Germany).

### 4.7. Subcellular Localization Assay

Subcellular localization of BkLiP1 in *N. benthamiana* was assessed using a binary vector pCNF3 containing an eYFP gene, and the full-length *BkLiP1* and BkLiP1 without SP were fused to the N-terminus of eYFP between the *Xba* I and *Bam*H I digestion sites. The plasmid of H_2_B-mCherry [83] was used as nucleus markers. All the constructs were transformed into *A. tumefaciens* GV3101 (Weidi Bio, Shanghai, China) using a heat shock method and infiltrated into *N. benthamiana* (4-week-old) leaves as previously described [15]. The empty pCNF3-eYFP vector was used as a control. At 2 days post infiltration, the *N. benthamiana* leaves were harvested and imaged under confocal laser scanning microscopy (CLSM; Leica Microsystems, TCS-SP8, Germany). For plasmolysis, *N. benthamiana* leaves were treated with 1 M NaCl to induce plasmolysis before fluorescence detection [15].

### 4.8. Cell Death, ROS Activity, and Callose Deposition Induction Assay

To detect the cell death, ROS activity, and callose deposition of the BkLiP1, each recombined plasmid was infiltrated into four leaves of *N. benthamiana* seedlings for expression as described above. The sequences of *BkLiP1* and *BkLiP1*^∆SP^ were cloned separately into the PVX vector through *Cla* I and *Sma* I digestion (TaKaRa, Dalian, China). The Bax and GFP were used as positive and negative controls, respectively. In addition, the pCNF3-BkLiP1-eYFP and pCNF3-BkLiP1^ΔSP^-eYFP vectors were also used (pCNF3-eYFP was used as negative control). The accumulation of ROS in the plant leaves of *N. benthamiana* was detected using a 3,3′-diaminobenzidine (DAB) solution as previously described [84]. To detect the deposition of callose, the *N. benthamiana* leaves were stained with aniline blue 48 h after infiltration as described previously [85]. According to previous research [86], trypan blue staining with the leaves of *N. benthamiana* was used to test the cell death in the later stages (two weeks) of transient expression. Transient protein expression in *N. benthamiana* was verified by Western blot using anti-FLAG antibodies (Smart lifescience, Changzhou, China) [57]. The leaf samples of *N. benthamiana* were collected 24 h after infiltration for nucleic acid extraction. All the experiments were repeated three times.

## Figures and Tables

**Figure 1 ijms-23-06066-f001:**
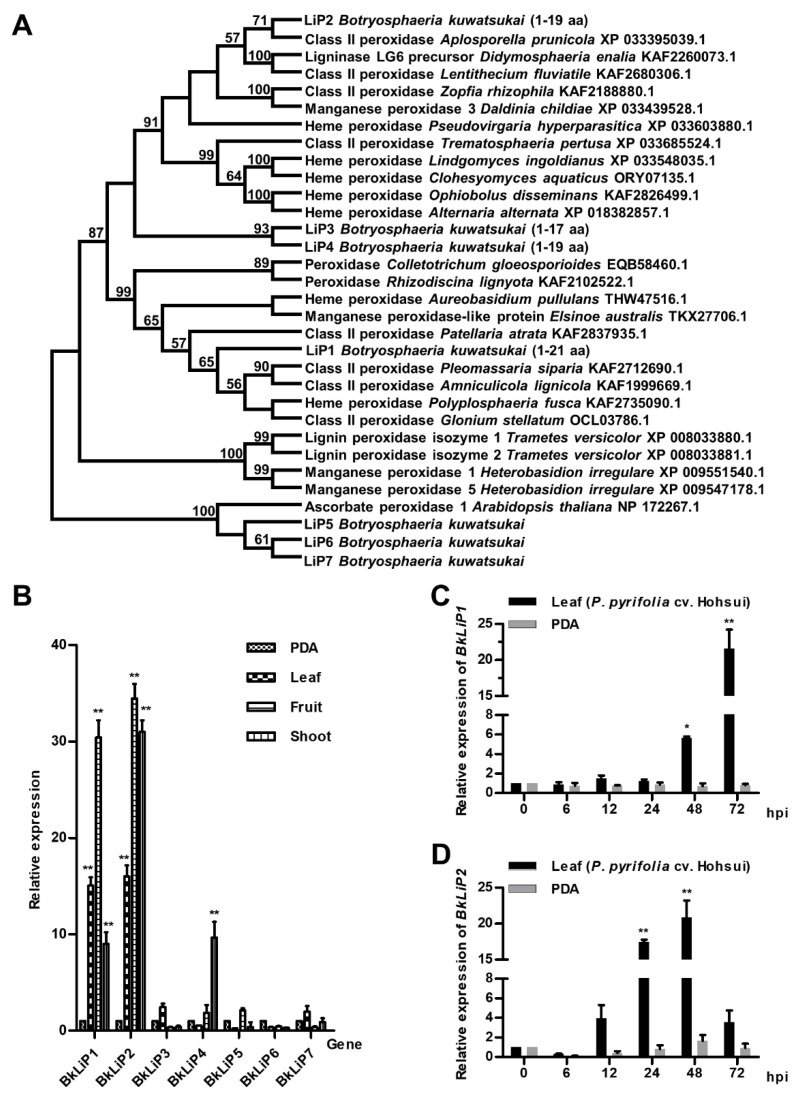
Phylogenetic analysis and induced expression of *lignin peroxidase* (*LiP*) genes in *Botryosphaeria kuwatsukai*. (**A**) The evolutionary relationship of LiP orthologs in *B. kuwatsukai* and other fungi determined with the maximum-likelihood algorithm. The statistical strengths were assessed by bootstrap with 1000 replicates. Bootstrap values are shown near the tree nodes, and signal peptide (SP) positions are in parentheses. (**B**) Relative levels of transcript accumulation of seven *BkLiP* genes determined by RT-qPCR in inoculated leaves (diamond stripe columns), fruits (horizontal stripe columns), shoots (vertical stripe columns) of pear (*Pyrus pyrifolia* cv. Hohsui), or on PDA (intersect stripe columns) at 25 °C for 3 days. (**C**) Relative levels of transcript accumulation of *BkLiP1* determined by RT-qPCR in inoculated pear leaves (black columns) or on PDA (gray columns) at 25 °C for 0–72 h. (**D**) Relative levels of transcript accumulation of *BkLiP2* determined by RT-qPCR in inoculated pear leaves (black columns) or on PDA (gray columns) at 25 °C for 0–72 h. The relative levels of transcripts were calculated using the comparative Ct method. The levels of 18S rRNA transcript of *B. kuwatsukai* were used to normalize different samples. Values were the means of three independent trials. Bars indicate ± SE. Asterisks at the top of the bars indicate statistical significance (* *p* < 0.05; ** *p* < 0.01).

**Figure 2 ijms-23-06066-f002:**
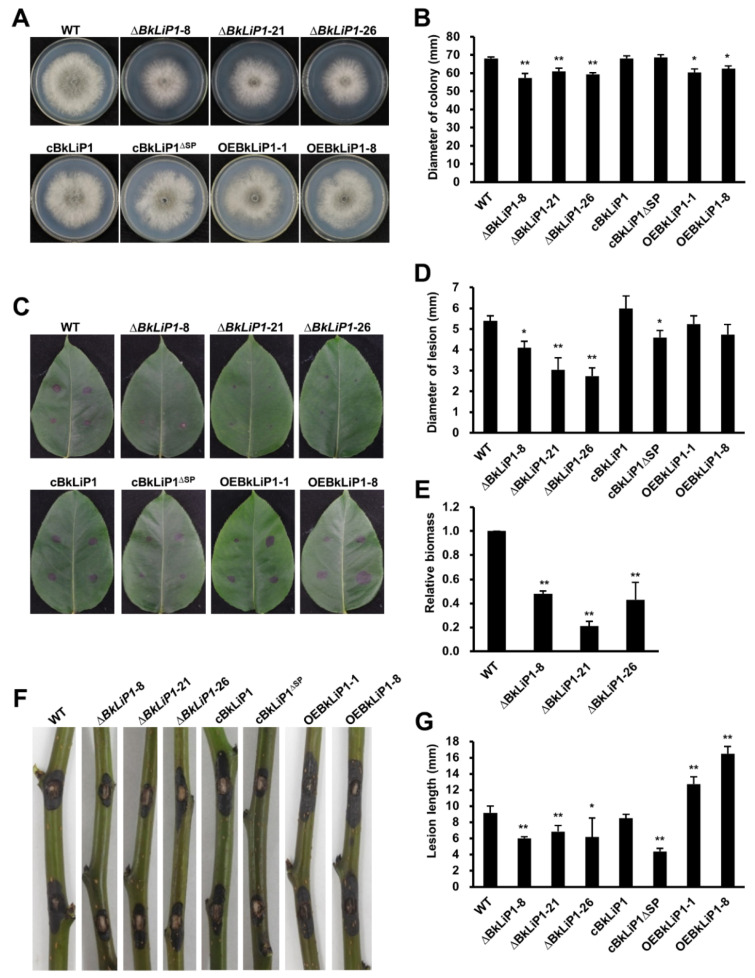
Growth assessment and virulence test of *B. kuwatsukai* and its derivative strains by knock-out or recovery of *BkLiP1*. (**A**,**B**) Morphologies and colony diameters as grown on PDA for 3 days at 28 °C for wild-type strain SZ2l62 (WT), the *BkLiP1* deletion mutants (∆*BkLiP1*-8, ∆*BkLiP1*-21, and ∆*BkLiP1*-26), the *BkLiP1* complemented transformants (cBkLiP1 and cBkLiP1^∆SP^), and the *BkLiP1* overexpression strains (OEBkLiP1-1 and OEBkLiP1-8). (**C**,**D**) Representative symptoms and lesion sizes on the detached pear (*P. pyrifolia* cv. Hohsui) leaves at 4 dpi induced by *BkLiP1* deletion mutants, complemented transformants, and overexpression strains. (**E**) Relative biomass of *B. kuwatsukai* in infected pear (*P. pyrifolia* cv. Hohsui) leaves at 4 dpi determined by qPCR. (**F**,**G**) Representative symptoms and lesion sizes on the detached pear (*P. pyrifolia* cv. Hohsui) shoots at 3 dpi induced by *BkLiP1* deletion mutants, complemented transformants, and overexpression strains. Virulence was evaluated based on lesion length. Mean and standard deviation were estimated with data of more than three (n > 3) independent biological replicates. Bars indicate ± SE. Asterisks at the top of the bars indicate statistical significance (* *p* < 0.05; ** *p* < 0.01).

**Figure 3 ijms-23-06066-f003:**
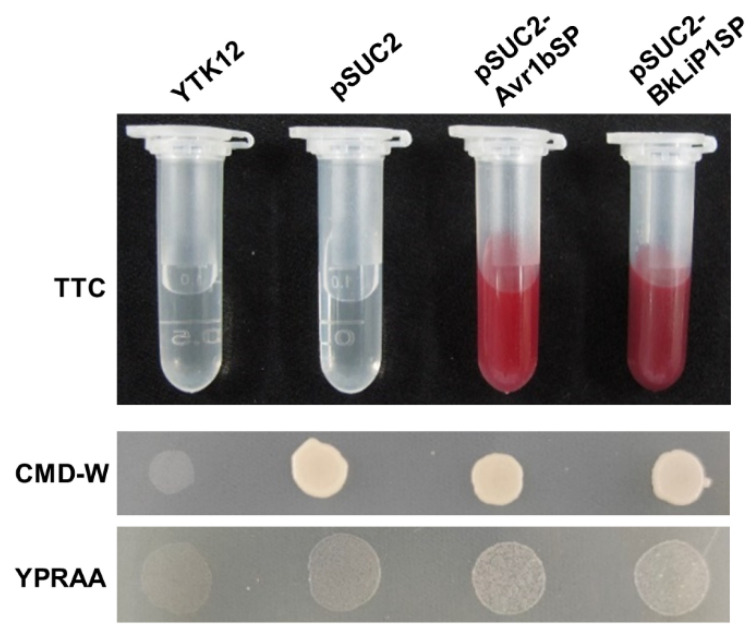
Secretion activity analysis of BkLiP1 in *B. kuwatsukai*. Invertase activity analysis in TTC medium and yeast growth on CMD-W or YPRAA plates. The yeast SUC2 mutant YTK12 and its transformants express the empty vector pSUC2 or vectors with the signal peptide from BkLiP1 and Avr1 (positive control).

**Figure 4 ijms-23-06066-f004:**
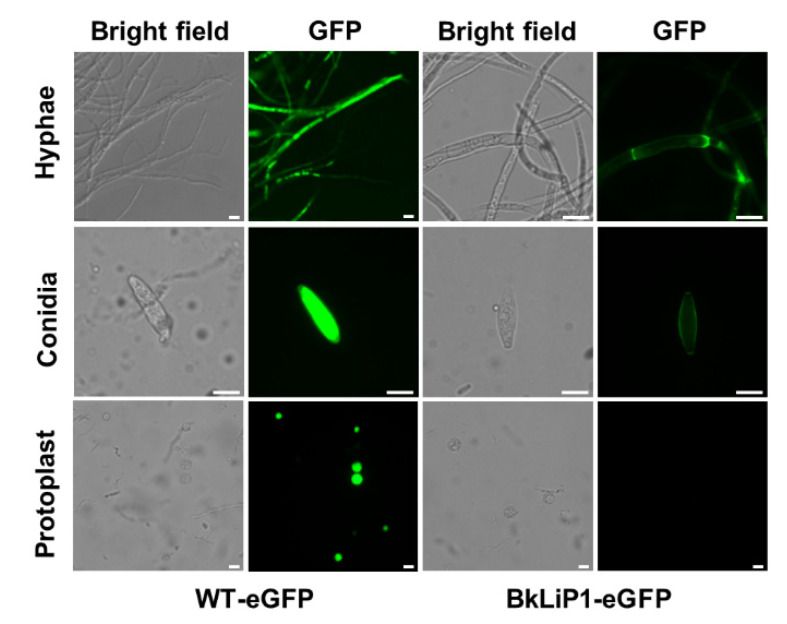
Localization analysis of BkLiP1 in *B. kuwatsukai*. Localization of eGFP and fusion protein BkLiP1-eGFP in the hyphae, conidia, and protoplast of *B. kuwatsukai* observed under a fluorescence microscope. Scale bar = 10 µm.

**Figure 5 ijms-23-06066-f005:**
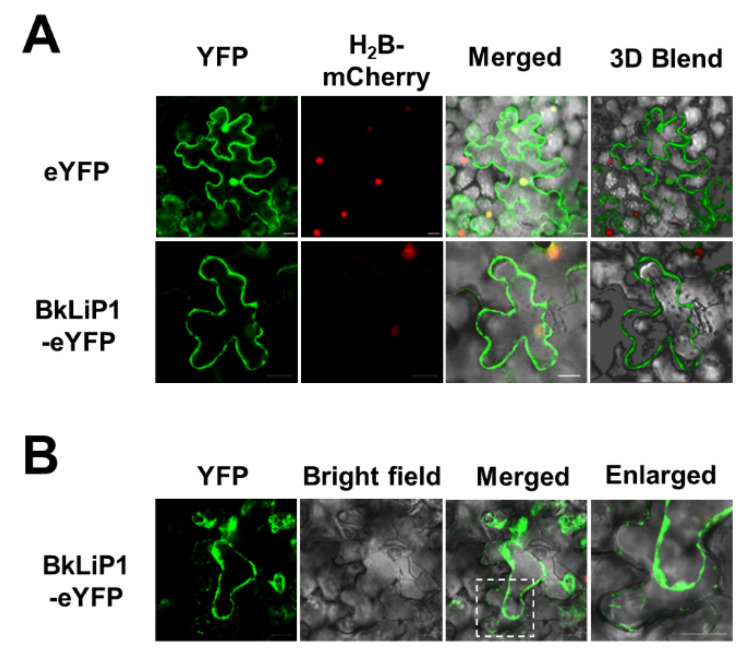
Subcellular localization analysis of BkLiP1 *N. benthamiana*. (**A**) Subcellular localization of BkLiP1 in epidermal cells of *N. benthamiana* leaves. The fungal protein BkLiP1 was expressed as fused to the N-terminus of eYFP. The fusion protein H_2_B-mCherry was used as a nucleus marker. The same imaging conditions were used in the three channels and 3D Blend projection. (**B**) Subcellular localization of BkLiP1 in epidermal cells of *N. benthamiana* leaves after plasmolysis. The enlarged section of each image was highlighted with a box. Images were acquired 2 days after agroinfiltration under confocal laser scanning microscopy. The images show maximum projections of 4 confocal images captured along the *z*-axis. Scale bar = 20 μm.

**Figure 6 ijms-23-06066-f006:**
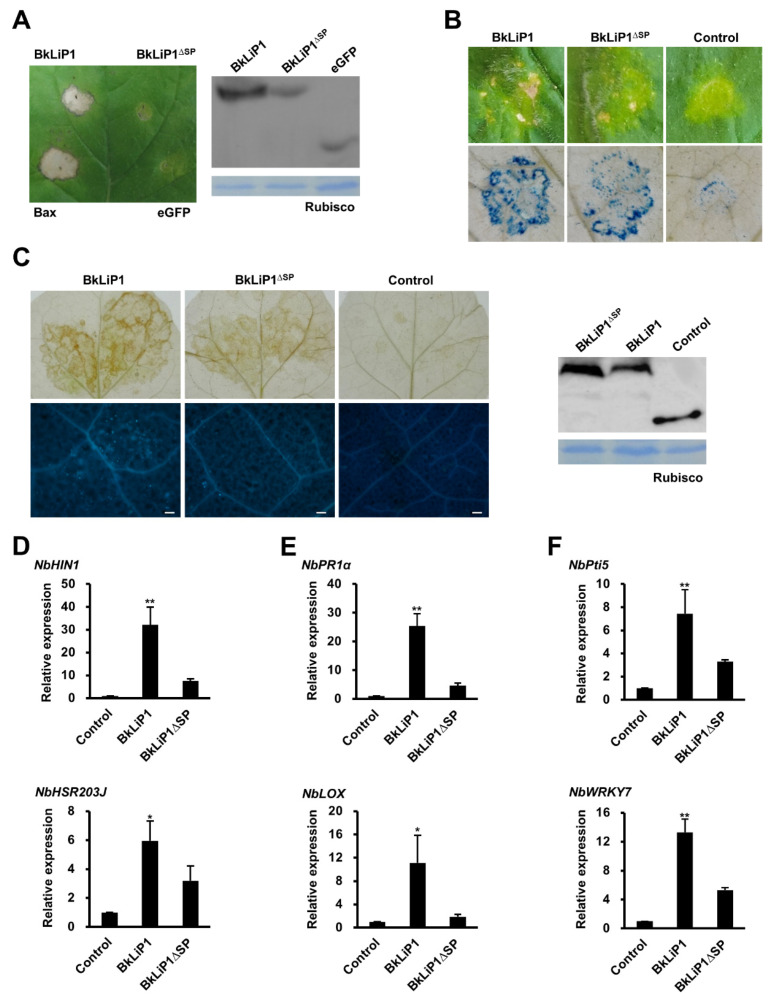
Immune responses triggered by BkLiP1 in *N. benthamiana*. (**A**) Programmed cell death triggered by transient expression of BkLiP1 in *N. benthamiana* leaves at 7 dpi infiltrated with *Agrobacterium tumefaciens* containing PVX-BAX, PVX-BkLiP1, PVX-BkLiP1^∆SP^, and PVX-GFP vector, respectively. Western blot analysis was conducted to confirm protein expression with a Flag tag protein in *N. benthamiana* leaves 48 h after infiltrated by *A. tumefaciens* cells. (**B**) Cell death in *N. benthamiana* leaves at 13 dpi infiltrated with *A. tumefaciens* containing pCNF3-BkLiP1, pCNF3-BkLiP1^∆SP^, and pCNF3-eYFP (control) vector, respectively. (**C**) Accumulation of reactive oxygen species (ROS, Up) and callose deposition (Down) in *N. benthamiana* at 2 dpi. Scale bar = 100 μm. Western blot analysis was conducted to confirm protein expression with a Flag tag protein in *N. benthamiana* leaves 48 h after being infiltrated by *A. tumefaciens* cells. (**D**–**F**) Relative expression of hypersensitive response (HR) marker genes (**D**), genes associated with hormone signaling pathways (**E**), and the microbe-associated molecular pattern (MAMP) triggered immunity (PTI) marker genes (**F**) in *N. benthamiana* after infiltrated by *A. tumefaciens* containing pCNF3-BkLiP1, pCNF3-BkLiP1^∆SP^, and pCNF3-eYFP (control), respectively. Total RNA was extracted and transcript levels were analyzed by RT-qPCR. *NbActin* was used as the internal reference gene. Three independent replicates were performed. Bars indicate ± SE. Asterisks at the top of the bars indicate statistical significance (* *p* < 0.05; ** *p* < 0.01).

## Data Availability

The data presented in this study are available in the article.

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
