# Peer review of "A Secreted Lignin Peroxidase Required for Fungal Growth and Virulence and Related to Plant Immune Response"

_ijms, 2022, doi:10.3390/ijms23116066_

Round 1

Reviewer 1 Report

The paper submitted for review brings new information regarding the virulence of B. kuwatsukai. I believe that the title of the work should be improved. There is too much information contained in it. In addition, the plan of the experience is not entirely clear to me. How many plants have been infested? The experiment concerns one species. Shouldn't the plant species be included in the title? Will the immune response be the same as in other plants?

Author Response

We are pleased to note the favorable comments of reviewers in their opening sentence. And we thank the reviewer very much for the evaluation on our manuscript and the positive comments.

We shortened the title, and designated it as “A Secreted Lignin Peroxidase Required for Fungal Growth and Virulence and Related to Plant Immune Response”.

Since Nicotiana benthamiana has been widely utilized in innumerable studies, similar to Arabidopsis thaliana as a model plant to characterize a gene function in vivo, we chose N. benthamiana seedlings to analyze the plant immune response in this study, and believe that the plant immune response characterized in N. benthamiana could be strongly similar to those in other plants.

Additionally, we improved the section of Materials and Methods with more detailed information about the seedlings involved.

Reviewer 2 Report

Dear Authors,
the manuscript submitted for review is very interesting, well prepared and I did not notice any factual errors. I believe that the experiments have been selected correctly, the materials and methods are well described, the results correctly and carefully presented. I would suggest adding information about the extent of Botryosphaeria Kuwatsukai infection in which the problem is, are there any specific, numerical data?
I would also suggest slightly modifying the title of the work. It is too long, the "and" repeats itself - I think that stylistically this title could be better formulated.

Author Response

We are pleased to note the favorable comments of reviewers in their opening sentence. And we thank the reviewer very much for the evaluation on our manuscript and the positive comments.

We shortened the title as “A Secreted Lignin Peroxidase Required for Fungal Growth and Virulence and Related to Plant Immune Response”, and supplied more information about the resulting problem of Botryosphaeria kuwatsukai infection.
